# Latent functional diversity may accelerate microbial community responses to temperature fluctuations

**Thomas P Smith[1]\*, Shorok Mombrikotb[1], Emma Ransome[1], Dimitrios - Georgios Kontopoulos[2,3], Samraat Pawar[1], Thomas Bell[1]**

[1]The Georgina Mace Centre for the Living Planet, Imperial College London, Ascot, United Kingdom; [2]LOEWE Centre for Translational Biodiversity Genomics, Frankfurt, Germany; [3]Senckenberg Research Institute, Frankfurt, Germany

**\*For correspondence:**
thomas.smith1@imperial.ac.uk

**Competing interest:** The authors declare that no competing interests exist.

**Abstract** How complex microbial communities respond to climatic fluctuations remains an open question. Due to their relatively short generation times and high functional diversity, microbial populations harbor great potential to respond as a community through a combination of strain-level phenotypic plasticity, adaptation, and species sorting. However, the relative importance of these mechanisms remains unclear. We conducted a laboratory experiment to investigate the degree to which bacterial communities can respond to changes in environmental temperature through a combination of phenotypic plasticity and species sorting alone. We grew replicate soil communities from a single location at six temperatures between 4°C and 50°C. We found that phylogenetically and functionally distinct communities emerge at each of these temperatures, with *K*-strategist taxa favored under cooler conditions and *r*-strategist taxa under warmer conditions. We show that this dynamic emergence of distinct communities across a wide range of temperatures (in essence, community-level adaptation) is driven by the resuscitation of latent functional diversity: the parent community harbors multiple strains pre-adapted to different temperatures that are able to 'switch on' at their preferred temperature without immigration or adaptation. Our findings suggest that microbial community function in nature is likely to respond rapidly to climatic temperature fluctuations through shifts in species composition by resuscitation of latent functional diversity.

## Editor's evaluation

This important study tests potential mechanisms for microbial community adaptation to temperature. Using elegant experiments, the authors convincingly show that selection on standing variation in the community, that is, species sorting, drives the community response to temperature. This article will be of interest to ecologists and microbiologists studying the impacts of global change on community and ecosystem processes.

## Introduction

Microbes are drivers of key ecosystem processes. They are tightly linked to the wider ecosystem as pathogens, mutualists, and food sources for higher trophic levels, and also play a central role in ecosystem-level nutrient cycling, and therefore, ultimately in global biogeochemical cycles. Temperature has a pervasive influence on microbial communities because of its direct impact on microbial physiology and fitness (*Oliverio et al., 2017*; *García et al., 2018*; *Smith et al., 2019*). There is therefore great interest in understanding how temperature fluctuations impact microbial community dynamics and how those impacts affect the wider ecosystem (*Bardgett et al., 2008*).

**eLife digest** Most ecosystems on Earth rely on dynamic communities of microorganisms which help to cycle nutrients in the environment. There is increasing concern that climate change may have a profound impact on these complex networks formed of large numbers of microbial species linked by intricate biochemical relationships.

Any species within a microbial community can acclimate to new temperatures by quickly tweaking their biological processes, for example by activating genes that are more suited to warmer conditions. Over time, a species may acclimate or adapt to new conditions. However, the community as a whole can also respond to these changes, and often much faster, by simply altering the abundance or presence of its members through a process known as species sorting. It remains unclear exactly how acclimation, adaptation and species sorting each contribute to the community's response to a temperature shift – an increasingly common scenario under global climate change.

To address this question, Smith et al. investigated how species sorting and acclimation may help whole soil bacterial communities to cope with lasting changes in temperature. To do so, soil samples from a single field site (and therefore featuring the same microbial community) were incubated for four weeks under six different temperatures. Genetic analyses revealed that, at the end of the experiments, distinct communities specific to a given temperature had emerged. They all differed in species composition and the types of biological functions they could perform.

Further experiments showed that each community had been taken over by strains of bacteria which grew best at the new temperature that they had been exposed to, including extreme warming scenarios never seen in their native environment. This suggests that these organisms were already present in the original community. They had persisted even under temperatures which were not optimal for them, acting as a slumbering ('latent') 'reservoir' of traits and functional abilities that allowed species sorting to produce distinct and functionally capable communities in each novel thermal environment. This suggests that species sorting could help bacterial communities to cope with dramatic changes in their thermal environment.

Smith et al.'s findings suggest that bacterial communities can cope with warming environments much better than has been previously thought. In the future, this work may help researchers to better predict how climate change could impact microbial community structure and functioning, and most crucially their contributions to the global carbon cycle.

Temperature varies at practically all biologically relevant timescales, from seconds (e.g., sun/shade), through daily and seasonal fluctuations, to longer-term changes, including anthropogenic climate warming and fluctuations over geological timescales. Whole microbial communities can respond to temperature changes over time and space through phenotypic (especially, physiological) plasticity (henceforth, 'acclimation'), as well as genetic adaptation in their component populations (*Bennett et al., 1990*; *Kishimoto et al., 2010*; *Blaby et al., 2012*; *Kontopoulos et al., 2020a*). Microbial thermal acclimation can occur relatively rapidly (timescales of minutes to days) through processes such as activation and up- or downregulation of particular genes and alteration of fatty acids used in building cell walls (*Suutari and Laakso, 1994*). Adaptation is a necessarily slower process (timescales of weeks or longer) occurring either through selection on standing genetic variation in the population or that arising through recombination and mutation (*Bennett et al., 1990*; *Padfield et al., 2016*; *Barton et al., 2020*).

In addition, a third key mechanism through which microbial communities can respond to changing temperatures is species sorting (*Leibold et al., 2004*; *Wu et al., 2018*): changes in community composition through species-level selection where taxa maladapted to a new temperature are replaced by those that are pre-adapted to it. This can happen either relatively rapidly through the resuscitation or suppression of taxa that are already present (*Lennon and Jones, 2011*; *Wisnoski and Lennon, 2021*), or more slowly through immigration-extinction dynamics driven by dispersal from the regional species pool (*Langenheder and Székely, 2011*; *Wu et al., 2018*). Resuscitation may be an important mechanism driving species sorting in microbial communities in particular because many microbial taxa have the capacity to form environment-resistant spores when conditions are unfavorable, and then rapidly activate metabolic pathways and resuscitate in favorable conditions. This effectively widens

their thermal niche to allow persistence in the face of temperature change (***Lennon and Jones, 2011***; ***Wisnoski and Lennon, 2021***). In order for rapid resuscitation of dormant taxa to allow species sorting to drive community-level adaptation, there must be a wide source pool of species to select from. Indeed, sequencing studies have revealed the presence of thousands of distinct microbial taxa in small environmental samples, most occurring at low abundance (***Lynch and Neufeld, 2015***; ***Sogin et al., 2006***; ***Thompson et al., 2017***). There is also strong evidence that bacteria are often found well outside of their thermal niche. For example, thermophilic taxa are often found in cold ocean beds and cool soils (***Marchant et al., 2008***; ***Hubert et al., 2009***; ***Zeigler, 2014***). Thus, a significant reservoir of latent microbial functional diversity may be commonly present for species sorting to act upon (***Lennon and Jones, 2011***; ***Wisnoski and Lennon, 2021***).

Understanding the relative importance of acclimation, adaptation, and species sorting in the assembly and turnover (succession) of microbial communities is key to determining the rate at which they can respond to different regimes of temperature fluctuations. For example, a combination of acclimation and species sorting through resuscitation would enable rapid responses to sudden temperature changes, relative to adaptation. A number of past studies have investigated responses of microbial community composition and functioning to temperature changes, showing that composition can respond rapidly to warming (***Allison and Martiny, 2008***; ***Aydogan et al., 2018***), often correlated with responses of ecosystem functioning (***Karhu et al., 2014***; ***Melillo et al., 2017***; ***Yu et al., 2018***). However, a mechanistic basis of these community-level responses remains elusive, both in terms of how individual taxa respond to changing temperatures in a community context and the relative importance of acclimation, adaptation, and species sorting. The community context of the responses of individual microbial populations is important because interactions between strains can constrain or accelerate acclimation as well as adaptive evolution (***Scheuerl et al., 2020***). Also, while the importance of species sorting in microbial communities per se has been studied (***Van der Gucht et al., 2007***; ***Langenheder and Székely, 2011***; ***Székely and Langenheder, 2014***), work on this issue in the context of environmental temperature is practically nonexistent.

A further consideration is whether differing temperature conditions, such as the frequency and magnitude of temperature fluctuations, may influence the life history strategies of the taxa in the community (***Gilchrist, 1995***; ***Basan et al., 2020***), which will in turn alter the relative importance of sorting, acclimation, and adaption. In order to identify the life history strategies of bacteria, we must quantify their phenotypic traits, such as growth rates and yield (***Malik et al., 2020***). Quantifying these traits can allow us to identify growth specialists ($r$-strategists) and carrying-capacity specialists ($K$-strategists) (***Marshall, 1986***), and thus test whether these strategies are differentially favored in different thermal environments. By identifying life history strategies, we can consider the ecosystem implications of any adaptation-, acclimation-, or sorting-driven changes in microbial communities (***Malik et al., 2020***).

Here, we investigate whether species sorting and latent functional diversity alone can influence the response of soil bacterial communities to changes in environmental temperature. To this end, we subject replicate communities, shielded from immigration, to a wide range of temperatures in the laboratory. In order to understand the mechanistic basis of observed community-level changes, we analyze the phylogenetic structure and functional traits of the resulting component taxa.

## Materials and methods

We performed a species-sorting experiment to investigate how microbial communities respond to shifts in temperature (***Figure 1***). After each community incubation at a given temperature, we estimated the thermal optimum ($T_{opt}$) for every isolated strain by measuring the thermal performance curve (TPC) of its maximal growth rate across several temperatures (***Figure 1D***). This allowed us to determine how strain-level thermal preferences and niche widths vary with community growth (isolation) temperature, and the presence of taxa pre-adapted to the new temperature. We also performed a phylogenetic analysis of the overall assemblage to identify whether deep evolutionary differences predict which taxa (and their associated traits) are favored by sorting at different temperatures. To quantify strain-level functional traits, we measured their available cellular metabolic energy (ATP), respiration rates, and biomass yield at population steady state (carrying capacity), which allowed us to identify $r$- vs. $K$-strategists as well as trade-offs between different strategies.

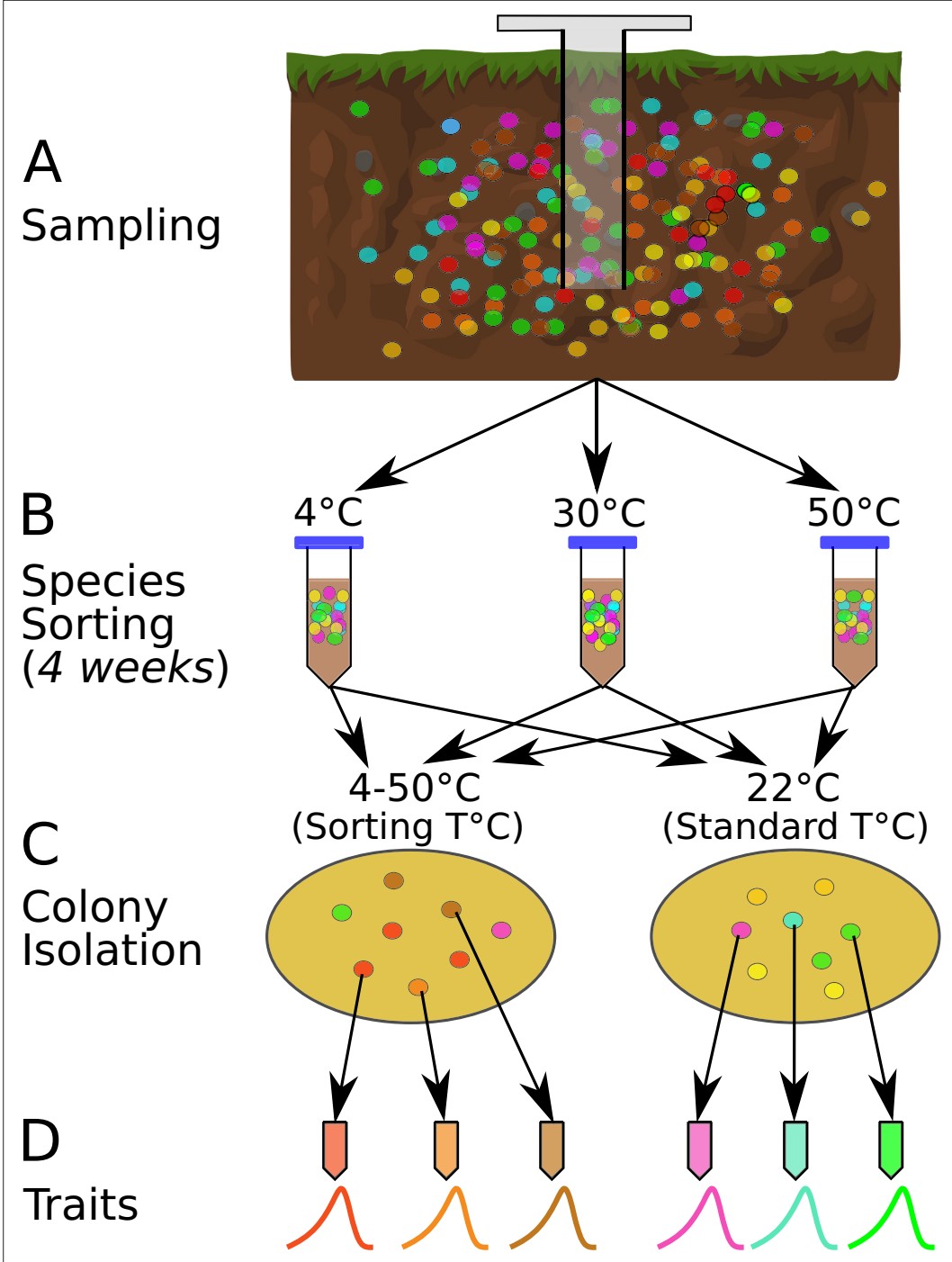

**Figure 1.** The species sorting experiment. (**A**) Different bacterial taxa (colored circles) sampled from the soil community. (**B**) Samples maintained at 4, 10, 21, 30, 40, and 50°C (only three temperatures shown for illustration), allowing species sorting for 4 weeks. (**C**) Soil washes from each core plated out onto agar and grown at both the sorting temperature and 22°C (standard temperature) to allow further species sorting and facilitate isolation (next step). (**D**) The six most abundant (morphologically different) colonies from each plate were picked, streaked, and isolated, and their physiological and life history traits measured. The curves represent each strain's unique unimodal response of growth rate to temperature.

## Species-sorting experiment

Soil cores were taken from a single site in Nash's Field (Silwood Park, Berkshire, UK, the site of a long-term field experiment [*Macdonald et al., 2015*]) in June 2016 (*Figure 1A*). Six cores were taken from the top 10 cm of soil, using a 1.5-cm-diameter sterile corer. Ambient soil temperature at the time of sampling was 19.4°C. The cores were maintained at different temperatures in the laboratory (4, 10, 21, 30, 40, and 50°C) for 4 weeks to allow species sorting to occur at those temperatures (*Figure 1B*). The soil was rehydrated periodically with sterile, deionized water during incubation. During this period, in each microcosm (incubated soil core), we expected some taxa would go extinct if the temperature was outside their thermal niche, and that survivors would acclimate to the new local thermal conditions. We also expected that the 4-week incubation period would be sufficient time for changes to species interactions due to changes in abundance or traits, and therefore that interaction-driven sorting would occur in addition to the immediate extinctions and acclimation. Because bacteria display higher growth rates at warmer temperatures (*Smith et al., 2019*), the different incubation conditions could result in differential generational turnover of species across the given timescale. However, we did not supplement the soil samples with any additional nutrients and thus expect any growth of bacteria during this time to be heavily restricted due to nutrient limitation. Therefore, environmental exclusion (elimination of taxa maladapted to the temperature conditions) was expected to be the dominant process affecting the bacterial taxa during this stage of the sorting experiment, rather than changes in abundances due to population growth. We then isolated bacterial strains by washing the soil with PBS, plating the soil wash onto R2 agar, and incubating the plates at both their 4-week incubation temperature treatments ('sorting temperature') and at 22°C ('standard temperature').

The sorting temperature allowed us to determine whether strains in each community tended to have thermal optima-matching experimental temperatures, while the standard temperature allowed us to determine whether a 4-week incubation resulted in a loss of taxa that were poorly adapted to 22°C. Appearance of strains with thermal optima matching the standard temperature would indicate incomplete species sorting because the 4-week treatment at temperatures higher or lower than 22°C had not eliminated (or at least suppressed) them.

The plates were incubated until bacterial colonies formed, of which we isolated a single colony from each of the six most abundant morphologically distinct colony types on each plate (*Figure 1C*). Additional species sorting likely occurred during this plating-based isolation because strains with the highest growth rates at each temperature would be the first to form visible colonies and be selected. The time frame for colony appearance on the agar plates differed between temperature treatments, ranging from (~10 days at 4°C to ~1.5 days at 50°C). Morphologically distinct colonies were isolated from each of the six sorting-temperature and six standard-temperature plates on R2 agar by streak-plating, before being frozen as glycerol stocks (*Figure 1*), which were later revived for trait measurements (see below). In total, 74 strains were isolated in this way.

### Taxonomic identification

16S rDNA sequences were used to identify the isolates. Raw sequences were first trimmed using Geneious 10.2.2 (https://www.geneious.com), and BLAST searches were then used to assign taxonomy to each trimmed sequence at the genus level. GenBank accession numbers of sequences are provided in Table 2.

## Quantifying physiological and life history traits

### Growth, respiration, and ATP content

We measured growth rate and respiration rate simultaneously across a range of temperatures for each isolate to construct its acute TPCs for these two traits. We henceforth denote the maximum growth rate across the temperature range by $\mu_{max}$, and the temperature at which this growth rate maximum occurs as $T_{opt}$ (optimal growth temperature or thermal optimum). The ATP content of the entire cell culture was also measured at the start and end of the growth assay. Strains were revived from glycerol stocks into fresh LB broth and incubated to carrying capacity at the temperature of the subsequent experiment. This growth to carrying capacity was an acclimation period, which typically took between 72 hr (warmest temperatures) to 500 hr (coldest temperature). Biomass abundance was determined by $OD_{600}$ – optical density measurements at 600 nm wavelength. Prior to each growth-respiration assay, the strains were diluted 1:100 in LB, pushing them into a short lag phase before exponential

growth started again (also tracked by $OD_{600}$ measurements). The exponentially growing cultures were subsequently centrifuged at 8000 rpm for 5 min to pellet the cells, which were then resuspended in fresh LB to obtain 400 µl culture at a final $OD_{600}$ of ~0.2–0.3. This yielded cells primed for immediate exponential growth without a lag phase. These cultures were serially diluted in LB (50% dilutions) three times, producing a range of starting densities of growing cells (four biological replicates per strain/ temperature combination). 100 µl subcultures of each replicate population were taken and $OD_{600}$ was tracked in a Synergy HT microplate reader (BioTek Instruments, USA) to ensure that cells were indeed in exponential growth. Initial ATP measurements were made using the BacTiter-Glo assay (see below for details) and cell counts were taken using a BD Accuri C6 flow cytometer (BD Biosciences, USA). Cells were then incubated with a MicroResp plate to capture cumulative respiration (see below for details of the MicroResp system) at the experimental temperature and allowed to continue growing for a short period of time (typically 3–4 hr). After growth, the MicroResp plate was read, and final cell count and ATP measurements taken.

We estimated average cell volumes and calculated the cellular carbon per cell from the flow cytometry cell diameter measurements using the relationship from *Romanova and Sazhin, 2010*:

$$\text{fgC cell}^{-1} = 133.754 V^{0.438}.$$

Multiplying this by the cell counts gives an estimate of the carbon biomass of the culture at the starting and ending points.

The difference between the initial biomass and biomass at the end of the experiment gives the total carbon sequestered through growth. Given an initial biomass ($C_0$) that grows over time ($t$) to reach a final biomass ($C_{tot}$), assuming the population is in exponential growth, the mass-specific growth rate (µ) is given by

$$\mu = \frac{\log(\frac{C_{tot}}{C_0})}{t}.$$

Respiration rates of cultures were measured during growth using the MicroResp system (*Campbell et al., 2003*). This is a colorimetric assay initially developed to measure $CO_2$ production from soil samples, which has since been used to measure respiration of bacterial cultures (*Lawrence et al., 2012*; *Foster and Bell, 2012*; *Rivett et al., 2017*). We calculate the biomass-specific respiration rate using an equation that accounts for changes in biomass of the growing cultures over time (*Smith et al., 2021*):

$$R = \frac{\mu R_{tot}}{C_0 e^{\mu t} - C_0}.$$

Here, $R_{tot}$ is the total mass of carbon produced according to the MicroResp measurements, $C_0$ is the initial population biomass, µ is the previously calculated growth rate, and $t$ is the experiment duration.

ATP content of the cultures was measured using the Promega BacTiter-Glo reagent, which produces luminescence in the presence of ATP, proportional to the concentration of ATP. 50 µl of culture (diluted 1:100) was incubated with 25 µl reagent. Luminescence was measured over a 6 min period to allow the reaction to develop completely, and measurements of luminescence recorded at the 0, 2, 4, and 6 min timepoints. The highest relative light unit (RLU) measurement for each culture was taken and used to calculate the quantity of ATP, using $\log(\text{nM ATP}) = 1.21 \cdot \log(\text{RLU}) - 4.69$, derived from a calibration curve. This was then normalized by the flow cytometry measurements to calculate the value of ATP/biomass.

## Thermal performance curves

To quantify TPCs of individual isolates, we fitted the Sharpe–Schoolfield model with the temperature of peak performance ($T_{pk}$) as an explicit parameter (*Schoolfield et al., 1981*; *Kontopoulos et al., 2020b*) to the experimentally derived temperature-dependent growth rate and respiration rates of each isolate:

$$B(T) = B_0 \frac{e^{\frac{-E}{k} \cdot \left(\frac{1}{T} - \frac{1}{T_{ref}}\right)}}{1 + \frac{E}{E_D - E} e^{\frac{E_D}{k} \left(\frac{1}{T_{pk}} - \frac{1}{T}\right)}}. \tag{1}$$

Here, $T$ is the temperature in Kelvin (K), $B$ is the biological rate (in this case, either growth rate, μ, or respiration rate, $R$), $B_0$ is the temperature-independent metabolic rate constant approximated at some (low) reference temperature $T_{ref}$, $E$ is the activation energy in electron volts (eV) (a measure of 'thermal sensitivity'), $k$ is the Boltzmann constant ($8.617 \times 10^{-5}$ eV K$^{-1}$), $T_{pk}$ is the temperature where the rate peaks, and $E_D$ is the deactivation energy, which determines the rate of decline in the biological rate beyond $T_{pk}$. We then calculated the peak performance (i.e., $R_{max}$ or $\mu_{max}$) by solving **Equation 1** for $T = T_{pk}$. This model was fitted to each dataset using a standard nonlinear least-squares procedure (**Smith et al., 2021**).

The $T_{pk}$ for growth rate was considered the optimum growth temperature (i.e., $T_{opt}$) for each isolate. Then, the operational niche width was calculated as the difference between $T_{opt}$ and the temperature below this value where $\mu_{max}$ ($B(T)$ in **Equation 1**) reached 50% of its maximum (i.e., $\mu_{max}$ at $T_{opt}$). This, a measure of an organism's thermal niche width relevant to typically experienced temperatures (**Pawar et al., 2016**; **Kontopoulos et al., 2020a**), was used as a quantification of the degree to which an isolate is a thermal generalist or specialist.

In most cases, $T_{opt}$ was derived directly from the Sharpe–Schoolfield flow cytometry growth rate fits. Four strains of *Streptomyces* were unsuitable for standard flow cytometry methods due to their formation of mycelial pellets (**van Veluw et al., 2012**). For these strains, growth rates derived from optical density measurements were used to estimate $T_{opt}$ instead.

## Trade-offs between traits

To understand the trade-offs and collinearities between different life history and physiological traits, we performed a principal components analysis (PCA), with optimum growth temperature ($T_{opt}$), niche width, peak growth rate ($\mu_{max}$), peak respiration rate ($R_{max}$), mean cellular ATP content (log-transformed), and carrying capacity (OD$_{600}$) as input variables (scaled to have mean = 0 and SD = 1).

All rate calculations, model fitting, and analyses were performed in R.

## Comparison to alternative datasets

We additionally investigated phylum-level life history strategy differences in two previously collated meta-analysis datasets as a comparison to our findings. **DeLong et al., 2010** compiled data on both active (growth phase) and passive (stationary phase) metabolic rates, as well as growth rates, across a range of bacteria (mainly from **Makarieva et al., 2005**), which were corrected to 20°C using an activation energy of 0.61 eV. We also investigated differences in the growth rates of bacteria compiled in **Smith et al., 2019**, which we temperature-corrected to 20°C here for comparison to the **DeLong et al., 2010** dataset, based on each strain's individual TPC parameters.

## Phylogenetic trait mapping

We used 16S sequences to build a phylogeny in order to investigate the evolution of thermal performance across the isolated bacterial taxa. Sequences were aligned to the SILVA 16S reference database using the SILVA Incremental Aligner (SINA) (**Pruesse et al., 2012**). From this alignment, 100 trees

**Table 1.** Details of time tree calibration nodes.
We constrained the time calibration of our RAxML tree based on estimated divergence times from TimeTree (**Kumar et al., 2017**).

| Taxa A | Taxa B | Min divergence time (MYA) | Max divergence time (MYA) |
|---|---|---|---|
| Bacteria | Archaea | 4290 | - |
| *Pseudomonas* | *Bacillus* | 3100 | 3254 |
| *Pseudomonas* | *Labrys* | 1053 | 3135 |
| *Pseudomonas* | *Collimonas* | 1053 | 3135 |
| *Collimonas* | *Variovorax* | 1271 | - |
| *Arthrobacter* | *Streptomyces* | 1420 | 1870 |
| *Arthrobacter* | *Bacillus* | 3100 | 3254 |
| *Bacillus* | *Brevibacillus* | 1734 | 2398 |

were inferred in RAxML (v8.1.1) using a GTR-gamma nucleotide substitution model. The tree with the highest log-likelihood was taken and time-calibrated using PLL-DPPDiv, which estimates divergence times using a Dirichlet Process Prior (*Heath et al., 2012*). DPPDiv requires a rooted phylogeny with the nodes in the correct order; however, RAxML by default produces an unrooted tree. Therefore, we included an archaeal sequence in our 16S alignment (*Methanospirillum hungatei*, RefSeq accession NR_074177) and used this as an outgroup in our RAxML run. This gives a tree rooted at the outgroup, which we checked for correct topology using TimeTree (*Kumar et al., 2017*) as a reference. We derived calibration nodes from TimeTree (*Kumar et al., 2017*) and performed two DPPDiv runs for 1 million generations each, sampling from the posterior distribution every 100 generations. We ensured that the two runs had converged by verifying that each parameter had an effective sample size above 200 and a potential scale reduction factor below 1.1. We summarized the output of DPPDiv into a single tree using the TreeAnnotator program implemented in BEAST (*Bouckaert et al., 2019*). We then dropped the outgroup tip to give a time-calibrated phylogeny of our bacterial 16S sequences only, which was used for further analysis. Details of calibration nodes used are given in *Table 1*.

To test whether there was evidence of evolution of $T_{opt}$, we calculated Pagel's $\lambda$ (*Pagel, 1999*), which quantifies the strength of phylogenetic signal – the degree to which shared evolutionary history has driven trait distributions at the tips of a phylogenetic tree. $\lambda = 0$ implies no phylogenetic signal, that is, the signal expected if variation in trait values is independent of the phylogeny. $\lambda = 1$ implies strong phylogenetic signal, that is, that the trait has evolved gradually along the phylogenetic tree (approximated as Brownian motion [BM]). Intermediate values ($0 < \lambda < 1$) imply deviation from the BM model, and may be observed for different reasons, such as constrained trait evolution due to stabilizing selection, and variation in evolutionary rate over time (e.g., due to episodes of rapid niche adaptation). Pagel's $\lambda$ requires that the trait be normally distributed. However, $T_{opt}$ in our dataset has a right-skewed distribution. Therefore, to test phylogenetic heritability we calculated $\lambda$ for $\log(T_{opt})$.

Blomberg's $K$ is another metric that is also widely used to infer phylogenetic heritability (*Blomberg et al., 2003*; *Münkemüller et al., 2012*). Blomberg's $K$ calculates the phylogenetic signal strength as the ratio of the mean squared error of the tip data and the mean squared error of the variance–covariance matrix of the given phylogeny under the assumption of BM (*Münkemüller et al., 2012*). $K = 1$ indicates taxa resembling each other as closely as would be expected under a BM model, $K < 1$ indicates less phylogenetic signal than expected under BM, and $K > 1$ indicates more phylogenetic signal than expected and thus a substantial degree of trait conservatism (*Blomberg et al., 2003*). Under a BM model of trait evolution, Pagel's $\lambda$ is expected to perform better than $K$, which may itself be better utilized for simulation studies (*Münkemüller et al., 2012*). Previous work suggests that $T_{pk}$ is likely to evolve in a BM manner in prokaryotes (*Kontopoulos et al., 2020a*), making $\lambda$ a more appropriate metric for these data than $K$. Furthermore, $\lambda$ is potentially more robust to incompletely resolved phylogenies and is therefore likely to provide a better measure than $K$ for ecological data in incomplete phylogenies (*Molina-Venegas and Rodríguez, 2017*). Therefore, we use $\lambda$ as likely the more appropriate metric for our data; however, for the sake of completeness, we also test for phylogenetic heritability using $K$.

We mapped the evolution of $T_{opt}$ onto our phylogeny using maximum likelihood to estimate the ancestral values at each internal node, assuming a BM model for trait evolution (an appropriate model, given the obtained $\lambda$ value). Where possible, we used $T_{opt}$ estimated directly from the Sharpe–Schoolfield fits. For six isolates whose growth was recorded at too few temperatures to fit the Sharpe–Schoolfield model, the temperature with the highest directly measured growth rate was taken as an estimate of $T_{opt}$.

The estimates of phylogenetic signal and the visualization of trait evolution were performed using tools from the R packages ape and phytools (*Revell, 2012*; *Revell and Freckleton, 2013*). The p-value for phylogenetic signal was based on a likelihood ratio test.

## Results

### Species sorting

In total, 74 strains of bacteria were isolated; 6 from each incubation temperature with matching sorting isolation temperature and 6 from each incubation temperature followed by a standard isolation temperature, with the exception of the 30°C sorting temperature regime, from which we

obtained eight isolates. Of these isolates, 60 could be reliably revived in liquid culture, from which 54 grew across a wide enough temperature range to produce enough data points for fitting the Sharpe–Schoolfield model (*Equation 1*). The 60 strains that could be revived were from 16 genera within three bacterial phyla (*Table 2*).

Isolates were in general well adapted to their sorting temperature (*Figure 2A*). A quadratic linear regression model fitted the data well (p<0.0001, shown in *Figure 2A*) and was preferred to a straight-line regression model (ANOVA, p<0.0001). The deviation from a simple linear response arises because the $T_{opt}s$ of isolates from the three lowest temperatures (4, 10, and 21°C) are significantly higher than their sorting and isolation temperature (*Figure 2A*). In comparison, the $T_{opt}s$ of standard temperature isolates were largely independent of the temperatures that their community had been previously grown at (*Figure 2B*), indicating that species sorting of the 4-week period had been incomplete, that is, strains maladapted to those temperature treatments had not been eliminated and were able to be resuscitated.

## Evolution of $T_{opt}$

$T_{opt}$ displays a strong signal of phylogenetic heritability, closely approximating a BM model of trait evolution (Pagel's $\lambda = 0.97$, p<0.001, $n = 60$), that is, closely related species have more similar $T_{opt}$ than random pairs of species. Qualitatively the same result was obtained using Blomberg's $K$ metric ($K = 0.71$, p<0.001, $n = 60$). The estimated ancestral states of $T_{opt}$ were mapped onto a phylogeny, where it can be seen that colder- or hotter-adapted strains tend to cluster together (*Figure 3A*). The inferred evolution of $T_{opt}$ through time indicates that a large amount of the trait space (cool to hot) is explored by Firmicutes, while Actinobacteria and Proteobacteria are constrained to a much narrower range of (relatively cool) optimal growth temperatures (*Figure 3B*).

## Functional traits and life history strategies

We investigated the level of association and trade-offs between different traits in the two major phyla isolated (Firmicutes and Proteobacteria) using PCA (*Figure 4A*). Growth specialists (copiotrophs, $r$ specialists) are expected to grow rapidly but wastefully, and therefore have high ATP content in combination with high growth rates, but low overall yield (carrying capacity). Yield specialists (oligo-trophs, $K$ specialists) are expected to grow more slowly but more efficiently, and should therefore display the opposite pattern, that is, relatively low growth rates and ATP content, but high yield. The first two principal components explained 60.1% of the cumulative variation in the data. $T_{opt}$, carrying capacity, and respiration rate showed greatest loading on the first principal component (PC1), while growth rate and niche width load most strongly on PC2. The Firmicutes and Proteobacteria phyla are partitioned in this space. The positive loadings onto PC2 of growth rate and ATP content versus the negative loading of carrying capacity suggest an $r$ vs. $K$ growth strategy trade-off; Proteobacteria have traits associated with a $K$-selected life history strategy while Firmicutes tend to have traits associated with an $r$-selected strategy. Furthermore, thermal niche width loads positively on PC2 along with growth rate and ATP content, implying that thermal generalism is not traded off against growth rate in these taxa; that is, no thermal generalist-specialist trade-off in growth rates.

To further understand the partitioning of taxa into these life history strategies, we investigated the differences in accessible cellular energy (ATP) content between these two phyla. We found that across the entire dataset (all replicate measurements across all temperatures), respiration rate and ATP content display a power–law relationship in both phyla (*Figure 4B*). While Firmicutes have generally higher ATP levels overall, they display a sublinear scaling relationship of ATP levels with respiration rate (scaling exponent = 0.60 ± 0.07, p<0.001, $R^2 = 0.13$, $n = 1722$). In comparison, while Proteobacteria have less standing ATP content on average, they show an approximately linear scaling relationship between ATP and respiration rate (scaling exponent = 0.99 ± 0.06, p<0.001, $R^2 = 0.59$, $n = 710$). This suggests that Proteobacteria are deriving ATP from aerobic respiration only, whereas Firmicutes may be utilizing alternative pathways.

Finally, to ask whether the higher growth rates and lower respiration rates of Firmicutes comparative to Proteobacteria was a phenomenon constrained to our small dataset, or whether it was a more general trend observed between the two phyla, we compared this to data compiled in two meta-analyses – *DeLong et al., 2010* and *Smith et al., 2019*. In the *DeLong et al., 2010* data,

**Table 2.** List of revivable strains and GenBank accession numbers.

Strain codes follow XX_YY_ZZ naming convention, where XX is the incubation temperature, YY is the isolation temperature, and ZZ is a numeric designator for the specific isolate. RT = room temperature (22°C, termed 'standard temperature'). All 16S sequences are archived on NCBI's GenBank with the accession numbers indicated.

| Strain | Accession no. | Phylum | Class | Order | Family | Genus |
|---|---|---|---|---|---|---|
| 04_04_02 | ON804144 | Proteobacteria | Gammaproteobacteria | Pseudomonadales | Pseudomonadaceae | Pseudomonas |
| 04_04_04 | ON804145 | Proteobacteria | Betaproteobacteria | Burkholderiales | Oxalobacteraceae | Collimonas |
| 04_04_05 | ON804146 | Firmicutes | Bacilli | Bacillales | Bacillaceae | Bacillus |
| 04_04_06 | ON804147 | Proteobacteria | Gammaproteobacteria | Pseudomonadales | Pseudomonadaceae | Pseudomonas |
| 04_RT_01 | ON804148 | Firmicutes | Bacilli | Bacillales | Bacillaceae | Bacillus |
| 04_RT_02 | ON804149 | Proteobacteria | Gammaproteobacteria | Pseudomonadales | Pseudomonadaceae | Pseudomonas |
| 04_RT_03 | ON804150 | Firmicutes | Bacilli | Bacillales | Bacillaceae | Bacillus |
| 04_RT_05 | ON804151 | Proteobacteria | Gammaproteobacteria | Pseudomonadales | Pseudomonadaceae | Pseudomonas |
| 10_10_06 | ON804152 | Actinobacteria | Actinobacteria | Micrococcales | Micrococcaceae | Arthrobacter |
| 10_RT_01 | ON804153 | Proteobacteria | Gammaproteobacteria | Pseudomonadales | Pseudomonadaceae | Pseudomonas |
| 10_RT_02 | ON804154 | Proteobacteria | Gammaproteobacteria | Pseudomonadales | Pseudomonadaceae | Pseudomonas |
| 10_RT_03 | ON804155 | Firmicutes | Bacilli | Bacillales | Bacillaceae | Bacillus |
| 21_21_01 | ON804156 | Firmicutes | Bacilli | Bacillales | Bacillaceae | Bacillus |
| 21_21_02 | ON804157 | Proteobacteria | Gammaproteobacteria | Pseudomonadales | Pseudomonadaceae | Pseudomonas |
| 21_21_04 | ON804158 | Proteobacteria | Betaproteobacteria | Burkholderiales | Oxalobacteraceae | Collimonas |
| 21_21_05 | ON804159 | Firmicutes | Bacilli | Bacillales | Bacillaceae | Bacillus |
| 21_21_06 | ON804160 | Proteobacteria | Gammaproteobacteria | Pseudomonadales | Pseudomonadaceae | Pseudomonas |
| 21_RT_01 | ON804161 | Proteobacteria | Betaproteobacteria | Burkholderiales | Oxalobacteraceae | Collimonas |
| 21_RT_02 | ON804162 | Firmicutes | Bacilli | Bacillales | Bacillaceae | Bacillus |
| 21_RT_03 | ON804163 | Firmicutes | Bacilli | Bacillales | Paenibacillaceae | Paenibacillus |
| 21_RT_04 | ON804164 | Proteobacteria | Gammaproteobacteria | Xanthomonadales | Rhodanobacteraceae | Dyella |
| 21_RT_05 | ON804165 | Actinobacteria | Actinobacteria | Corynebacteriales | Nocardiaceae | Nocardia |
| 21_RT_06 | ON804166 | Firmicutes | Bacilli | Bacillales | Bacillaceae | Bacillus |
| 30_30_01 | ON804167 | Actinobacteria | Actinobacteria | Streptomycetales | Streptomycetaceae | Streptomyces |
| 30_30_02 | ON804168 | Firmicutes | Bacilli | Bacillales | Bacillaceae | Bacillus |
| 30_30_03 | ON804169 | Actinobacteria | Actinobacteria | Streptomycetales | Streptomycetaceae | Streptomyces |
| 30_30_04 | ON804170 | Firmicutes | Bacilli | Bacillales | Bacillaceae | Bacillus |
| 30_30_05 | ON804171 | Firmicutes | Bacilli | Bacillales | Bacillaceae | Bacillus |

*Table 2 continued*

| Strain | Accession no. | Phylum | Class | Order | Family | Genus |
|---|---|---|---|---|---|---|
| 30_30_06 | ON804172 | Actinobacteria | Actinobacteria | Streptomycetales | Streptomycetaceae | *Streptomyces* |
| 30_30_07 | ON804173 | Actinobacteria | Actinobacteria | Streptomycetales | Streptomycetaceae | *Streptomyces* |
| 30_30_08 | ON804174 | Proteobacteria | Alphaproteobacteria | Rhizobiales | Xanthobacteraceae | *Labrys* |
| 30_RT_01 | ON804175 | Proteobacteria | Betaproteobacteria | Burkholderiales | Comamonadaceae | *Variovorax* |
| 30_RT_02 | ON804176 | Proteobacteria | Betaproteobacteria | Burkholderiales | Comamonadaceae | *Variovorax* |
| 30_RT_03 | ON804177 | Firmicutes | Bacilli | Bacillales | Bacillaceae | *Bacillus* |
| 30_RT_04 | ON804178 | Proteobacteria | Gammaproteobacteria | Xanthomonadales | Rhodanobacteraceae | *Dyella* |
| 30_RT_05 | ON804179 | Proteobacteria | Betaproteobacteria | Burkholderiales | Comamonadaceae | *Variovorax* |
| 30_RT_06 | ON804180 | Firmicutes | Bacilli | Bacillales | Bacillaceae | *Bacillus* |
| 40_40_01 | ON804181 | Firmicutes | Bacilli | Bacillales | Bacillaceae | *Bacillus* |
| 40_40_02 | ON804182 | Firmicutes | Bacilli | Bacillales | Paenibacillaceae | *Cohnella* |
| 40_40_03 | ON804183 | Firmicutes | Bacilli | Bacillales | Bacillaceae | *Bacillus* |
| 40_40_04 | ON804184 | Firmicutes | Bacilli | Bacillales | Planococcaceae | *Rummeliibacillus* |
| 40_40_05 | ON804185 | Firmicutes | Bacilli | Bacillales | Paenibacillaceae | *Cohnella* |
| 40_40_06 | ON804186 | Firmicutes | Bacilli | Bacillales | Planococcaceae | *Viridibacillus* |
| 40_RT_01 | ON804187 | Firmicutes | Bacilli | Bacillales | Bacillaceae | *Bacillus* |
| 40_RT_02 | ON804188 | Firmicutes | Bacilli | Bacillales | Bacillaceae | *Bacillus* |
| 40_RT_03 | ON804189 | Firmicutes | Bacilli | Bacillales | Planococcaceae | *Viridibacillus* |
| 40_RT_04 | ON804190 | Firmicutes | Bacilli | Bacillales | Planococcaceae | *Viridibacillus* |
| 40_RT_05 | ON804191 | Firmicutes | Bacilli | Bacillales | Bacillaceae | *Bacillus* |
| 40_RT_06 | ON804192 | Firmicutes | Bacilli | Bacillales | Bacillaceae | *Bacillus* |
| 50_50_01 | ON804193 | Firmicutes | Bacilli | Bacillales | Planococcaceae | *Rummeliibacillus* |
| 50_50_02 | ON804194 | Firmicutes | Bacilli | Bacillales | Bacillaceae | *Anoxybacillus* |
| 50_50_03 | ON804195 | Firmicutes | Bacilli | Bacillales | Paenibacillaceae | *Brevibacillus* |
| 50_50_04 | ON804196 | Firmicutes | Bacilli | Bacillales | Paenibacillaceae | *Brevibacillus* |
| 50_50_05 | ON804197 | Firmicutes | Bacilli | Bacillales | Bacillaceae | *Bacillus* |
| 50_50_06 | ON804198 | Firmicutes | Bacilli | Bacillales | Bacillaceae | *Anoxybacillus* |
| 50_RT_01 | ON804199 | Firmicutes | Bacilli | Bacillales | Bacillaceae | *Bacillus* |
| 50_RT_02 | ON804200 | Firmicutes | Bacilli | Bacillales | Bacillaceae | *Bacillus* |
| 50_RT_03 | ON804201 | Firmicutes | Bacilli | Bacillales | Bacillaceae | *Bacillus* |
| 50_RT_04 | ON804202 | Firmicutes | Bacilli | Bacillales | Bacillaceae | *Bacillus* |

*Table 2 continued*

| Strain | Accession no. | Phylum | Class | Order | Family | Genus |
|--------|--------------|--------|-------|-------|--------|-------|
| 50_RT_06 | ON804203 | Firmicutes | Bacilli | Bacillales | Bacillaceae | *Bacillus* |

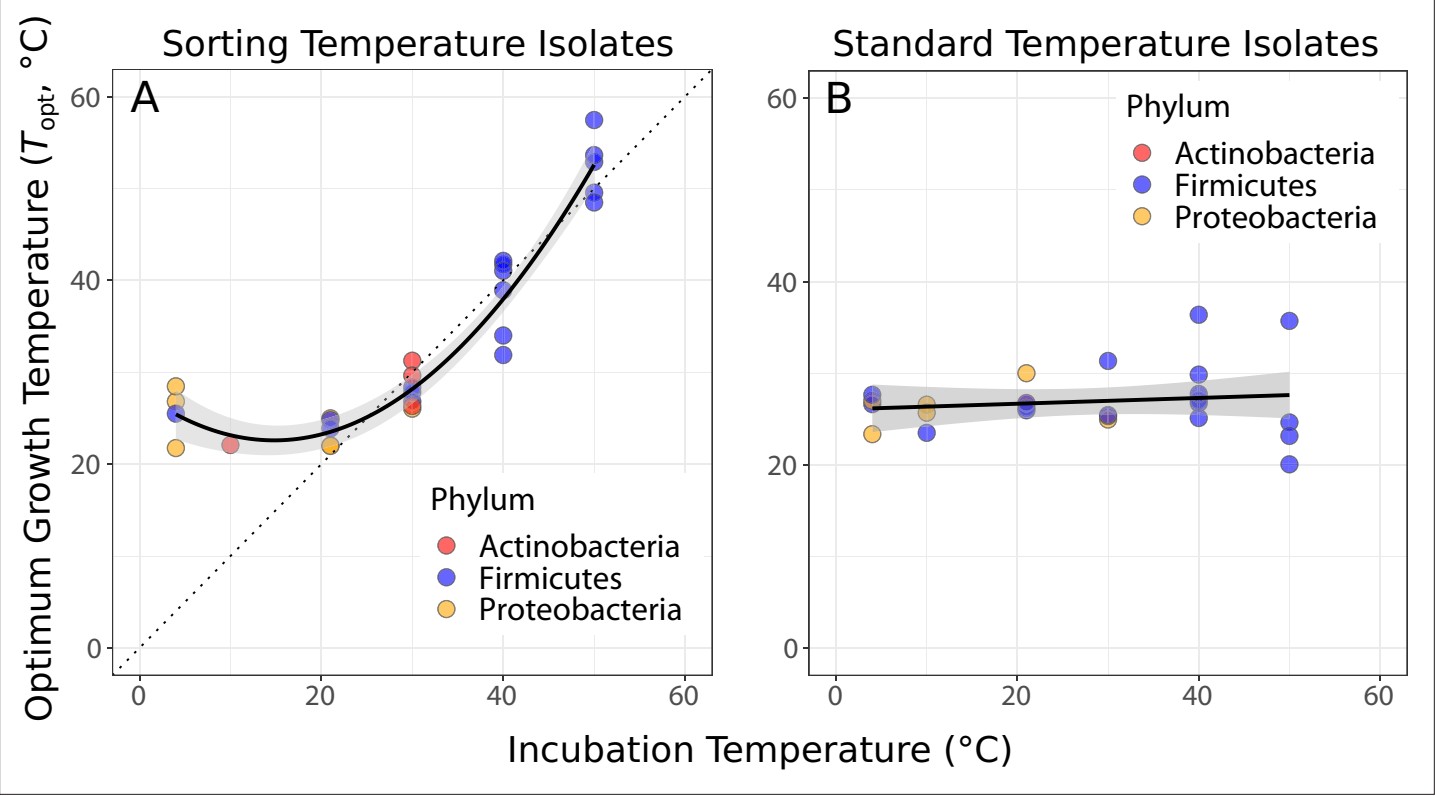

**Figure 2.** Species sorting of soil bacteria driven by temperature change. (**A**) Thermal optima of growth rate closely match sorting temperature for the isolates from those temperatures (black line: quadratic linear regression, p<0.0001, $R^2$ = 0.94, *n* = 28). Note that the prediction bounds at three lowest temperatures do not include the 1:1 (dashed) line. (**B**) No significant association between incubation temperature and thermal optima for standard temperature isolates (simple linear regression, p = 0.488, $R^2$ = 0.02, *n* = 26). These results show that species sorting can act upon latent diversity to select for isolates adapted to different temperature conditions (**A**), but that isolates maladapted to the sorting conditions can re-emerge (be resuscitated) under the appropriate conditions (**B**).

Proteobacteria have higher active and passive metabolic rates than Firmicutes (active rates Wilcoxon rank-sum test p=0.0017, *n* = 39; passive rates Wilcoxon rank-sum test p=0.0098, *n* = 108, **Figure 5A**), consistent with our findings; however, there is no significant difference between the growth rates of the two phyla in these data (Wilcoxon rank-sum test p=0.66, *n* = 31, **Figure 5B**). By comparison, the **Smith et al., 2019** dataset does show a significant difference between the growth rates of these phyla, with Firmicutes on average higher than Proteobacteria (Wilcoxon rank-sum test p=0.00035, *n* = 135, **Figure 5C**). We also compared the distribution of $T_{opt}$ for both phyla in the data from **Smith et al., 2019** and find that Proteobacteria account for much more of the low-temperature strains, while Firmicutes are more associated with high temperatures (**Figure 5D**), which is consistent with our temperature isolation findings here (**Figure 2**).

## Discussion

Here, using a novel species-sorting experiment, we have studied the extent to which species sorting and acclimation can influence the responses of soil bacterial communities to temperature change. We find that when replicate soil bacterial communities sampled from a temperate environment are subjected to a wide range of temperatures for 4 weeks, in microcosms where immigration is not possible, strains with thermal preferences matching the local conditions emerge consistently. The strong correspondence between strain-level optimal growth temperatures and isolation temperatures (**Figure 2A**) indicates that a pool of taxa with disparate thermal physiologies, including those maladapted to the ambient thermal conditions, persisted in the parent community. This result is reinforced by fact that the $T_{opt}$s of standard temperature isolates were largely independent of the temperatures that their community had been previously grown at (**Figure 2B**), indicating that strains

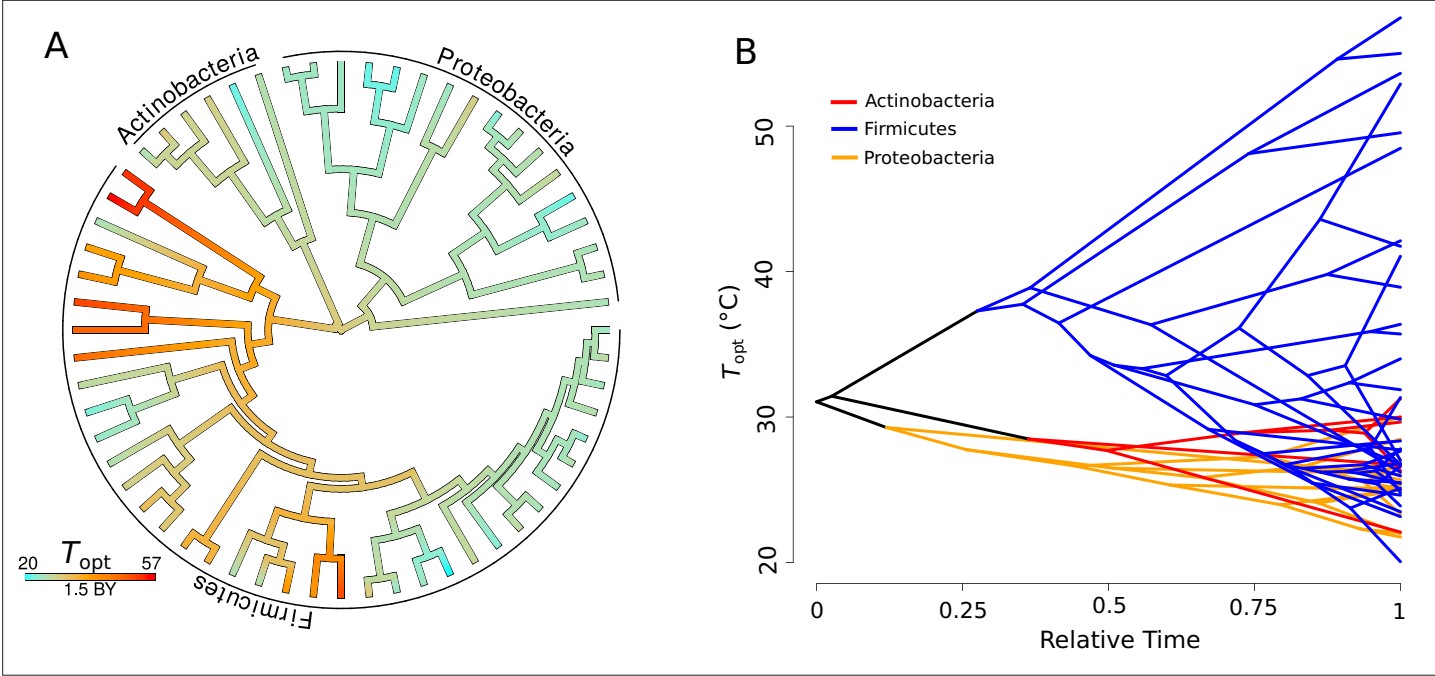

**Figure 3.** Evolution of $T_{opt}$. (**A**) Ancestral trait reconstruction of $T_{opt}$ visualized on a tree, from lower temperatures in cyan, to higher temperatures in red, with time given in billions of years (BY). All of the higher temperature (40–50°C) isolates belong to the phylum Firmicutes. (**B**) Projection of the phylogenetic tree into the $T_{opt}$ trait space (y-axis), over relative time (x-axis) since divergence from the root. The clades representative of each phylum are colored on the projection (Actinobacteria, red; Firmicutes, blue; Proteobacteria, yellow).

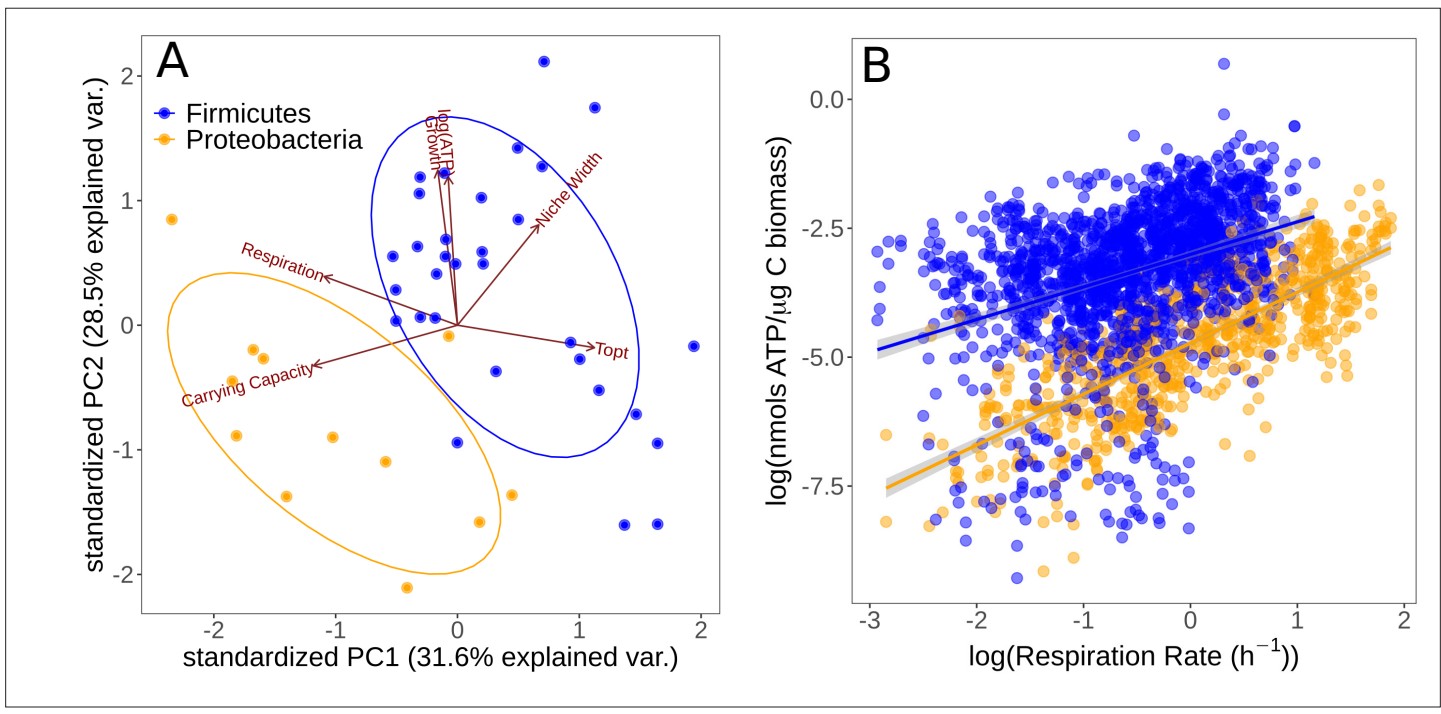

**Figure 4.** Partitioning of growth strategies between phyla. (**A**) Principal components analysis (PCA) on life history traits, colored by phylum. Relative to each other, Firmicutes (blue) tend to be $r$ specialists, Proteobacteria (orange) tend to be $K$ specialists. (**B**) ATP content of cultures is associated with the respiration rate. Firmicutes show a sublinear scaling relationship of ATP with respiration rate (scaling exponent = 0.60 ± 0.07), while Proteobacteria display an approximately linear scaling relationship (scaling exponent = 0.99 ± 0.06). The same color scheme is shared by both sub-plots.

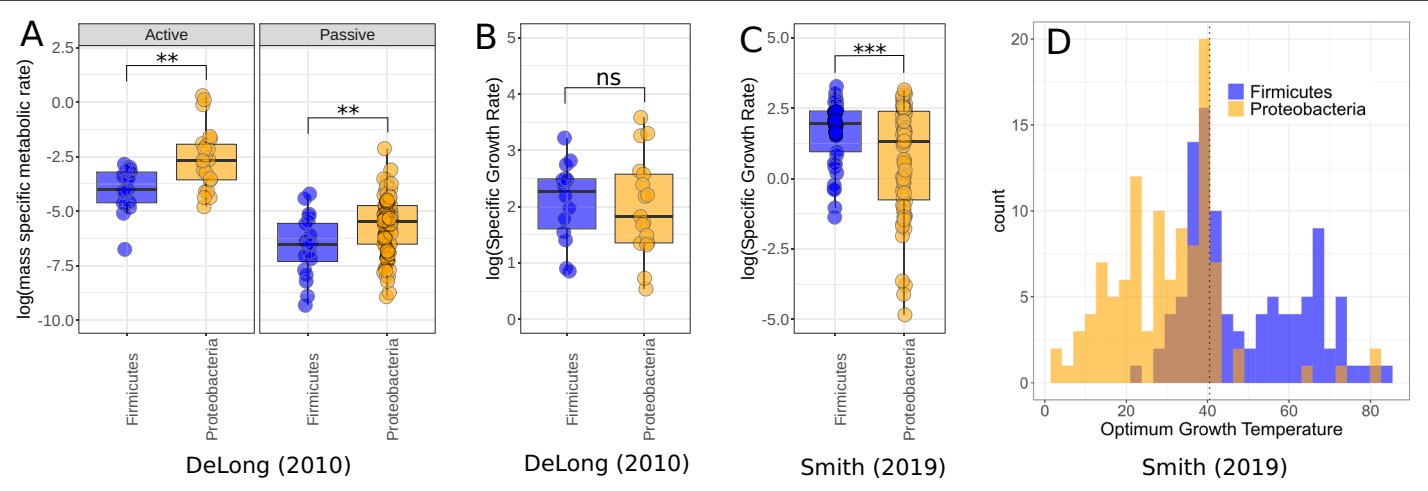

**Figure 5.** Comparison of Firmicutes and Proteobacteria in meta-analysis datasets. (**A**) Dataset used by *DeLong et al., 2010* shows significantly higher active (*n* = 39) and passive (*n* = 108) metabolic rates for Proteobacteria than Firmicutes. Significance determined by Wilcoxon rank-sum tests – ns, p≥0.05; *p<0.05; **p<0.01; ***p<0.001. (**B**) The growth rate data used by *DeLong et al., 2010* shows no significant difference between the phyla (*n* = 31). (**C**) The growth rate data from *Smith et al., 2019* does show significantly increased growth rates for Firmicutes over Proteobacteria however (*n* = 135). (**D**) Distribution of Firmicutes and Proteobacteria $T_{opt}$ from *Smith et al., 2019*. Proteobacteria account for a large proportion of the low-temperature strains, while Firmicutes dominate the high temperatures. Dotted line marks 40.5°C, a cut-off between mesophiles and thermophiles (*Smith et al., 2019*).

maladapted to that temperature had not been eliminated and were able to be resuscitated. Therefore, we conclude that most 'sorting' occurred during the isolation step of our experiment rather than during the 4-week incubation period – the thermal optima of taxa isolated reflects the isolation conditions.

While a 4-week period is arguably too short for mutation- or recombination-driven thermal adaptation in environmental samples (as a significant degree of generational turnover is required [*Bennett et al., 1990*; *Lenski, 2017*; *Chase et al., 2021*]), it is worth considering the possibility that some of the community-level emergence of thermally adapted strains could have been driven by rapid evolution through selection on standing trait variation. Indeed, stochastic mapping of thermal physiological traits on the prokaryotic tree of life has shown that $T_{opt}$ evolves relatively rapidly compared to other traits such as niche width or activation energy (thermal sensitivity) (*Kontopoulos et al., 2020a*). This is consistent with adaptive evolution experiments, which have shown that bacteria as well as archaea can rapidly adapt to new temperatures by shifting their $T_{opt}$ (*Bennett et al., 1990*; *Kishimoto et al., 2010*; *Blaby et al., 2012*; *Smith et al., 2019*). The molecular mechanisms underlying such rapid evolution are still being investigated, but structural changes to enzymes that alter their melting temperatures appear to be a key mechanism when adaptation to relatively high temperatures is called for (*Pucci and Rooman, 2017*). While determining whether such mechanisms can be operationalized over the duration of our sorting experiment was beyond the scope of our study, this is still arguably a very short time frame for significant shifts in thermal optima due to selection on standing variation alone. Furthermore, the communities that remained after 4 weeks of growth at the six temperatures consisted of taxonomically distinct sets of strains, and the $T_{opt}$s of the overall set of taxa exhibited a significant phylogenetic signature (*Figure 3*). This indicates that the observed systematic differences in $T_{opt}$ across the temperature-specific communities were driven by species sorting on preexisting physiological variation across strains rather than thermal adaptation of single strains. Overall, we therefore conclude that species sorting played a dominant role in determining the response of the parent community to abrupt changes in temperature, in the absence of immigration, and with negligible adaptation.

We also detected systematic turnover in functional traits that likely underpin the change in thermal optima with species sorting. There were differences in the taxa isolated at different temperatures, with more Proteobacteria at lower temperatures and more Firmicutes at higher temperatures (all $T_{opt} > 35°C$ were Firmicutes). Furthermore, these phyla were partitioned in the *r-K* and thermal

generalism-specialism trait spaces (*Figure 4*). Proteobacteria were found to be relatively $K$-selected thermal specialists and Firmicutes relatively $r$-selected thermal generalists. These findings are inconsistent with a generalist-specialist trade-off in which increasing thermal niche width is proposed to inevitably incur a metabolic cost, reducing maximal growth rates (*Huey and Hertz, 1984*; *Angilletta, 2009*). As with our findings, recent work on phytoplankton thermal performance traits also failed to detect a generalist-specialist trade-off (*Kontopoulos et al., 2020a*), questioning its universality in microbes. Since the existence of such a trade-off plays a key role in life history theory, there would be value in further experiments to confirm the generality of this finding.

The increased growth rates and lower respiration rates of Firmicutes relative to Proteobacteria found here are also largely consistent with datasets from meta-analyses of bacterial rates (*Figure 5*). Additionally, previously reported values for cellular ATP content have generally been found to be higher for Firmicutes than Proteobacteria with more than tenfold greater intracellular ATP content reported for *Bacillus* versus *Pseudomonas* strains (*Hattori et al., 2003*), some of the major representatives of Firmicutes and Proteobacteria in this experiment, respectively. This suggests that these phyla tend to allocate resources to growth and respiration in fundamentally different ways. One explanation for these seemingly divergent strategies may be found in Firmicutes deriving extra energy though fermentation pathways. There is a mechanistic trade-off between growth rate and yield whereby bacteria may increase their rate of ATP production by supplementing aerobic respiration with fermentation (*Pfeiffer et al., 2001*). Fermentation pathways increase the rate of ATP production but result in lower total yield, allowing populations to reach higher growth rates but lower carrying capacity from the same resource input. This is consistent with the apparent $r$ vs. $K$ selection trade-off observed in our results. The differences in scaling relationship between ATP content and respiration rate may provide further evidence of differences in metabolic pathways utilized. Across Proteobacteria, ATP content has a scaling exponent of approximately 1 with respiration rate, indicating that these strains are deriving ATP solely from aerobic respiration (*Figure 4B*). The fact that Firmicutes have a lower scaling exponent ($0.60 \pm 0.07$), that is, that they are generating higher levels of ATP than expected at lower rates of respiration, may indicate that they derive ATP from alternative pathways alongside aerobic respiration. These differences in metabolic strategies reflect underlying differences in the efficiency of growth, that is, carbon use efficiency (CUE), between these taxa (*Smith et al., 2021*). Moreover, CUE varies systematically with temperature in a phylogenetically structured manner (*Pold et al., 2020*; *Smith et al., 2021*). Thus, community turnover due to temperature change is likely to have a profound impact on community-level functional traits, such as CUE (*Domeignoz-Horta et al., 2020*).

In contrast to the strong association between $T_{opt}$ and incubation temperature in the sorting temperature isolates, we did not observe any similar relationships in the standard temperature isolates, where mesophiles were consistently recovered regardless of prior incubation conditions. This indicates that species sorting was incomplete (in that maladapted taxa were not driven extinct), implying that bacterial communities can be resilient to temperature change at the community level. Taxa suited to different temperatures are able to 'switch on' as conditions become suitable, allowing community-level functional plasticity due to the latent functional diversity present within communities. Although mesophiles were recovered from all incubation temperatures in our standard temperature experiment, there was the same taxonomic bias as seen in the sorting temperature isolates – more Firmicutes were recovered from higher temperatures. This is probably a reflection of the propensity of Firmicutes to form endospores and remain dormant until conditions are favorable, upon which they invest resources into rapid growth to gain a competitive advantage over other taxa, consistent with our life history trait findings of $r$-specialism in Firmicutes (*Lennon and Jones, 2011*). In comparison, the Proteobacteria in this experiment were generally more suited to oligotrophic environments (e.g., Collimonas; *Leveau et al., 2010*), where constituent species are expected to present low growth rate and high carrying capacity ($K$ specialists; *Fierer et al., 2007*), as well as increased respiration (*Keiblinger et al., 2010*). This idea is supported by the observation that we isolated the strains from sandy, acidic soil (i.e., oligotrophic) (*Fornara et al., 2013*), and sequencing studies revealed that Proteobacteria are the most abundant phylum (*Macdonald et al., 2015*). We do not suggest that this adoption of $r$ vs. $K$ strategy is general to all Firmicutes and Proteobacteria. Indeed, meta-analysis reveals little consistency in the phyla associated with copiotrophy or oligotrophy (*Ho et al., 2017*). Nor do we suggest that warming is likely to result in selection for Firmicutes over Proteobacteria – community temperature responses are not likely to be consistent at coarse phylogenetic levels (*Oliverio et al.,*

*2017*). However, the results presented here are consistent with phylum-specific traits for the majority of our isolates when compared to each other.

Patterns of microbial community succession in nature are driven by the differences in growth strategy between taxa that we report here. Studies have revealed taxonomic groups associated with different stages of microbial succession, with patterns broadly consistent across timescales of days (*Noll et al., 2005*; *Shrestha et al., 2007*; *Rui et al., 2009*) and years (*Nemergut et al., 2007*; *Banning et al., 2011*) and even over thousands and tens of thousands of years, as revealed through sequencing of soil sediments (*Jangid et al., 2013*). Generally, across these studies, the phyla Firmicutes and Bacteroidetes are associated with early succession, while other phyla such as Actinobacteria and Acidobacteria are more abundant at later stages of succession. Proteobacteria are less consistent at the phylum level, with Alphaproteobacteria associated with late succession, Betaproteobacteria associated with early succession and Gammaproteobacteria variously associated with different stages of succession in different studies. Isolated taxa reveal a strong association between early succession and high growth rates (*Shrestha et al., 2007*) as well as rRNA operon copy numbers, a key determinant of bacterial growth rate (*Klappenbach et al., 2000*). The $K$-selected taxa may therefore be thought of as general constituents of soil, associated with standard low turnover of carbon, while the $r$-selected taxa may be seen as more opportunistic from their involvement in early succession. Indeed, signatures of community-level differences in $r$- vs. $K$-selection have been observed in microbial communities at different successional stages (*Pascual-García and Bell, 2020*). Fluctuating temperatures may therefore drive repeated assembly dynamics via sorting on latent microbial diversity, leading to functional community changes through time. However, the frequency and magnitude of temperature fluctuations may also influence the life history strategies of the taxa in the community (*Gilchrist, 1995*; *Basan et al., 2020*).

Although we report patterns broadly consistent with previous findings at the phylum level, bacteria isolated from the environment will always represent only a small, incomplete subset of the overall diversity of the natural community. Previous 16S sequencing of the field site sampled here has revealed Proteobacteria to be the most abundant phylum, followed by Verrucomicrobia, Acidobacteria, Actinobacteria, and Firmicutes, respectively (*Macdonald et al., 2015*). That the majority of our isolates are from the Firmicutes and that we isolated no Acidobacteria or Verrucomicrobia, despite their expected relative abundances in these soils, is not surprising. Firmicutes are consistently overrepresented in culture libraries (*Schloss et al., 2016*; *Floyd et al., 2005*), while most members of the Acidobacteria and Verrucomicrobia are notoriously difficult to reliably culture (*Kielak et al., 2010*; *Kalam et al., 2020*). Therefore, caution should be taken when interpreting community responses from culture-based studies like ours.

In summary, we have found that resuscitation of latent functional diversity driven by phenotypically plastic responses of single taxa to temperature change can allow whole bacterial communities to track dramatic changes in temperature. Community function is expected to be driven by interactions between the most abundant taxa (*Rivett and Bell, 2018*) and therefore changes in the abundance of taxa with temperature variation are likely to drive profound changes in overall community functioning (mediated by community-level variation in traits such as CUE). In particular, $r$- vs. $K$-selection is likely to vary with temperature change at the community level, from daily to seasonal successional trajectories, driven by species sorting. Furthermore, climate change is expected to lead to increased temperature fluctuations (*Vasseur et al., 2014*), both in magnitude and frequency. This may potentially lead to more frequent species sorting effects over short timescales, further driving changes in community composition through time. Overall, these results show that latent diversity in thermal physiology combined with temperature induced species sorting is likely to facilitate the responses of microbial community structure and functioning to climatic fluctuations.

## Acknowledgements

TPS was supported by a BBSRC DTP scholarship (BB/J014575/1). TB and SP were funded by NERC grants NE/M020843/1 and NE/S000348/1.

## Additional information

### Funding

| Funder | Grant reference number | Author |
|---|---|---|
| Biotechnology and Biological Sciences Research Council | BB/J014575/1 | Thomas P Smith |
| Natural Environment Research Council | NE/M020843/1 | Samraat Pawar Thomas Bell |
| Natural Environment Research Council | NE/S000348/1 | Thomas P Smith Samraat Pawar Thomas Bell |

The funders had no role in study design, data collection and interpretation, or the decision to submit the work for publication.

### Author contributions

Thomas P Smith, Conceptualization, Resources, Data curation, Software, Formal analysis, Validation, Investigation, Visualization, Methodology, Writing - original draft, Project administration, Writing - review and editing; Shorok Mombrikotb, Investigation, Methodology, Writing - review and editing; Emma Ransome, Data curation, Investigation, Writing - review and editing; Dimitrios - Georgios Kontopoulos, Software, Formal analysis, Methodology, Writing - review and editing; Samraat Pawar, Conceptualization, Supervision, Funding acquisition, Project administration, Writing - review and editing; Thomas Bell, Conceptualization, Supervision, Funding acquisition, Methodology, Project administration, Writing - review and editing

### Author ORCIDs

Thomas P Smith http://orcid.org/0000-0002-4038-9722
Dimitrios - Georgios Kontopoulos http://orcid.org/0000-0002-5082-1929
Samraat Pawar http://orcid.org/0000-0001-8375-5684
Thomas Bell http://orcid.org/0000-0002-2615-3932

### Decision letter and Author response

Decision letter https://doi.org/10.7554/eLife.80867.sa1
Author response https://doi.org/10.7554/eLife.80867.sa2

## Additional files

### Supplementary files

• Transparent reporting form

### Data availability

16S sequences have been deposited in NCBI GenBank under accession codes ON804144:ON804203. Data and code to reproduce all analyses in this manuscript are provided on GitHub: https://github.com/smithtp/latent-diversity, (copy archived at swh:1:rev:d58dfefa598c6f701c0e7eb1da289aa9fc06c27b).

The following datasets were generated:

| Author(s) | Year | Dataset title | Dataset URL | Database and Identifier |
|---|---|---|---|---|
| Smith TP, Mombrikotb S, Ransome E, Kontopoulos D, Pawar S, Bell T | 2022 | Latent functional diversity may accelerate microbial community responses to temperature fluctuations | https://dx.doi.org/10.5061/dryad.f1vhhmh0g | Dryad Digital Repository, 10.5061/dryad.f1vhhmh0g |

*Continued*

| Author(s) | Year | Dataset title | Dataset URL | Database and Identifier |
|---|---|---|---|---|
| Smith TP, Mombrikotb S, Ransome E, Kontopoulos D, Pawar S, Bell T | 2022 | Pseudomonas sp. strain 04_04_02 16S ribosomal RNA gene, partial sequence | https://www.ncbi.nlm. nih.gov/search/all/? term=ON804144 | NCBI, ON804144 |
| Smith TP, Mombrikotb S, Ransome E, Kontopoulos D, Pawar S, Bell T | 2022 | Bacillus sp. (in: Bacteria) strain 50_RT_06 16S ribosomal RNA gene, partial sequence | https://www.ncbi.nlm. nih.gov/search/all/? term=ON804203 | NCBI, ON804203 |

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
