## [Editor Report]

This important study tests potential mechanisms for microbial community adaptation to temperature. Using elegant experiments, the authors convincingly show that selection on standing variation in the community, that is, species sorting, drives the community response to temperature. This article will be of interest to ecologists and microbiologists studying the impacts of global change on community and ecosystem processes.

---

## [Decision Letter]

**Decision letter after peer review:**

Thank you for submitting your article "Latent functional diversity may accelerate microbial community responses to temperature fluctuations" for consideration by *eLife*. Your article has been reviewed by 2 peer reviewers, and the evaluation has been overseen by a Reviewing Editor and Meredith Schuman as the Senior Editor. The reviewers have opted to remain anonymous.

Essential revisions:

1) Please expand the discussion of limitations to cultivation for understanding soil processes. What is the potential impact of cultivation bias on the interpretation of the study, considering e.g. impacts of growth temperatures and media (see reviewer comments)? Can you draw on data which allow you to estimate the magnitude of cultivation bias (see comments from Reviewer 2)?

2) Please explain the considerations leading to the experimental timeframe, and discuss impacts on the study's conclusions (fixed timeframe rather than fixed number of generations).

3) Please either revise the phylogenetic trait mapping analysis according to Reviewer 2's suggestions, or else explain and defend the current approach.

*Reviewer #1 (Recommendations for the authors):*

Overall this is a very good paper. Thank you to the authors for doing this work and reporting the results so clearly and carefully.

*Reviewer #2 (Recommendations for the authors):*

Strains were isolated by culture on R2 agar. How might this affect the subset of taxa that you have extracted from the community? Do you have any data on whole-community composition (for example by 16S gene barcoding) that you could use to evaluate how biased or incomplete your culture-based estimates of community composition might be?

For phylogenetic trait mapping, 16S sequences were aligned pairwise in MAFFT. I recommend alignment against a reference database such as SILVA using an algorithm such as PyNAST or a k-mer-based approach. Pairwise alignment of 16S sequences functions extremely poorly due to the hypervariable regions present in this gene. This could affect estimates of divergence times and tree topology.

---

## [Author Response]

Essential revisions:1) Please expand the discussion of limitations to cultivation for understanding soil processes. What is the potential impact of cultivation bias on the interpretation of the study, considering e.g. impacts of growth temperatures and media (see reviewer comments)? Can you draw on data which allow you to estimate the magnitude of cultivation bias (see comments from Reviewer 2)?

We have expanded our discussion of the limitations of culture-based studies and drawn upon previous work to illustrate the biases inherent in culture-based studies in lines 475-485 (detailed in our response to Reviewer 2’s comment #1, below).

2) Please explain the considerations leading to the experimental timeframe, and discuss impacts on the study's conclusions (fixed timeframe rather than fixed number of generations).

We have explained our choice of experimental timeframe in our response and argue that this is unlikely to have affected our findings. We have now clarified our manuscript to address this point in lines 130-136.

3) Please either revise the phylogenetic trait mapping analysis according to Reviewer 2's suggestions, or else explain and defend the current approach.

We have revised the phylogenetic analysis according to Reviewer 2’s suggestions and obtain qualitatively the same result as previously. We have incorporated this new methodology into our workflow and have updated the methods and Results sections of our manuscript to reflect this (see our response to Reviewer 2’s comment #2, below).

Reviewer #2 (Recommendations for the authors):Strains were isolated by culture on R2 agar. How might this affect the subset of taxa that you have extracted from the community? Do you have any data on whole-community composition (for example by 16S gene barcoding) that you could use to evaluate how biased or incomplete your culture-based estimates of community composition might be?

Indeed, culture-based studies result in the isolation of only a small, incomplete subset of a community, with many taxa over- and under-represented compared to their natural abundance. In particular, *Firmicutes* (the Phylum from which we cultured most isolates) tend to be over-represented in culture libraries [Schloss 2016; Floyd 2005], whilst many other potentially important community constituents remain unculturable.

We do not have data on whole community composition of the samples taken here, however the overall field site where samples were collected from has previously been profiled by 16S sequencing, and we noted in our manuscript previously that *Proteobacteria* were the most abundant phylum in these sequencing data (line 446).

We do agree with the reviewer that this is an important point, and we had not previously discussed this in enough detail. We have added a new section to the discussion comparing our findings to previous community composition work and reiterating the limitations of culture-based studies (lines 475-485):

“Although we report patterns broadly consistent with previous findings at the phylum-level, bacteria isolated from the environment will always represent only a small, incomplete subset of the overall diversity of the natural community. Previous 16S sequencing of the field site sampled here has revealed *Proteobacteria* to be the most abundant phylum, followed by *Verrucomicrobia*, *Acidobacteria*, *Actinobacteria* and *Firmicutes* respectively [Macdonald 2015]. That the majority of our isolates are from the *Firmicutes* and that we isolated no *Acidobacteria* or *Verrucomicrobia*, despite their expected relative abundances in these soils, is not surprising. *Firmicutes* are consistently over-represented in culture libraries [Schloss 2016; Floyd 2005], whilst most members of the *Acidobacteria* and *Verrucomicrobia* are notoriously difficult to reliably culture [Kielak 2010; Kalam 2020]. Therefore, caution should be taken when interpreting community responses from culture-based studies like ours.”.

For phylogenetic trait mapping, 16S sequences were aligned pairwise in MAFFT. I recommend alignment against a reference database such as SILVA using an algorithm such as PyNAST or a k-mer-based approach. Pairwise alignment of 16S sequences functions extremely poorly due to the hypervariable regions present in this gene. This could affect estimates of divergence times and tree topology.

We thank the reviewer for this helpful suggestion to improve the robustness of our phylogenetic trait mapping. As suggested, we have now aligned our sequences to the SILVA 16S database using the SILVA Incremental Aligner (SINA) [Pruesse 2012]. SINA uses a *k*-mer-based approach to alignment and is the program used to align the rRNA gene databases provided by the SILVA project itself.

Using this new alignment, we rebuilt our phylogeny, re-ran our analyses and re-generated the relevant figures (Figures 3A and 3B). We find qualitatively the same results as previously – strong phylogenetic heritability of *T*_opt_. We have updated our Results section with the slight changes in λ and *K* metrics for phylogenetic signal obtained with this new alignment (lines 305 and 307).

We have updated our methods section to reflect this change in methodology (lines 233-234):

“Sequences were aligned to the SILVA 16S reference database using the SILVA Incremental Aligner (SINA) [Pruesse 2012]”.

References

1. Smith TP, Thomas TJH, Garcia-Carreras B, Sal S, Yvon-Durocher G, Bell T, *et al.* Community-level respiration of prokaryotic microbes may rise with global warming. *Nature Communications*. 2019;10:5124. doi:10.1038/s41467-019-13109-1.

2. Macdonald CA, Crawley MJ, Wright DJ, Kuczynski J, Robinson L, Knight R, *et al.* Identifying qualitative effects of different grazing types on below-ground communities and function in a long-term field experiment. *Environmental Microbiology.* 2015;17(3):841–854. doi:10.1111/1462-2920.12539

3. Schloss PD, Girard RA, Martin T, Edwards J, Thrash JC. Status of the archaeal and bacterial census: An update. *mBio*. 2016;7(3):1–10. doi:10.1128/mBio.00201-16.

4. Floyd MM, Tang J, Kane M, Emerson D. Captured diversity in a culture collection: Case study of the geographic and habitat distributions of environmental isolates held at the American Type Culture Collection. *Applied and Environmental Microbiology*. 2005;71(6):2813–2823. doi:10.1128/AEM.71.6.2813-2823.2005

5. Kielak A, Rodrigues JLM, Kuramae EE, Chain PSG, Van Veen JA, Kowalchuk GA. Phylogenetic and metagenomic analysis of Verrucomicrobia in former agricultural grassland soil. *FEMS Microbiology Ecology*. 2010;71(1):23–33. doi:10.1111/j.1574-6941.2009.00785.x

6. Kalam S, Basu A, Ahmad I, Sayyed RZ, El-Enshasy HA, Dailin DJ, *et al.* Recent Understanding of Soil Acidobacteria and Their Ecological Significance: A Critical Review. *Frontiers in Microbiology*. 2020;11(October). doi:10.3389/fmicb.2020.580024.

7. Pruesse E, Peplies J, Glockner FO. SINA: Accurate high-throughput multiple sequence alignment of ribosomal RNA genes. *Bioinformatics*. 2012;28(14):1823–1829. doi:10.1093/bioinformatics/bts252.